

# Genotypic analyses and virulence characterization of *Glaesserella parasuis* isolates from Taiwan

Wei-Hao Lin[1,2], Hsing-Chun Shih[2], Chuen-Fu Lin[3], Cheng-Yao Yang[4], Chao-Nan Lin[1,2] and Ming-Tang Chiou[1,2,5]

[1] Department of Veterinary Medicine, College of Veterinary Medicine, National Pingtung University of Science and Technology, Pingtung, Taiwan
[2] Animal Disease Diagnostic Center, College of Veterinary Medicine, National Pingtung University of Science and Technology, Pingtung, Taiwan
[3] Department of Veterinary Medicine, College of Veterinary Medicine, National Chiayi University, Chiayi, Taiwan
[4] Graduate Institute of Veterinary Pathobiology, College of Veterinary Medicine, National Chung Hsing University, Taichung, Taiwan
[5] Research Center for Animal Biologics, National Pingtung University of Science and Technology, Pingtung, Taiwan

Corresponding authors
Chao-Nan Lin,
cnlin6@mail.npust.edu.tw
Ming-Tang Chiou,
mtchiou@mail.npust.edu.tw

## ABSTRACT

**Background:** *Glaesserella* (*Haemophilus*) *parasuis* (*G. parasuis*) causes severe economic losses in the swine industry. Multiple *G. parasuis* strains can exist in single animals. Typing techniques are required for identifying *G. parasuis* isolates. Different strains within a serovar display varying virulence. Enterobacterial repetitive intergenic consensus polymerase chain reaction (ERIC-PCR) can assess the heterogeneity. The group 1 virulence-associated trimeric autotransporters (vtaA) gene is an indicator of virulence. The aim of this study was to characterize Taiwanese *G. parasuis* isolates via molecular serotyping, vtaA PCR and ERIC-PCR.

**Methods:** One hundred and forty-five strains were collected between November 2013 and March 2017 in Taiwan and further examined by molecular serotyping, vtaA PCR and ERIC-PCR.

**Results:** The dendrogram revealed heterogeneous genetic diversity within many clusters. Partial correlation between the ERIC-PCR clusters of different strains, serovars and lesion patterns was observed. Twelve herds (8.3%) infected with more than one strain. Group 1 vtaA positive rate reached 98.6%.

**Discussion:** This study showed the high genetic diversity of *G. parasuis* in Taiwan by a high discriminatory capability of ERIC-PCR. Group 1 vtaA commonly exists in *G. parasuis* isolates and may play important roles in the pathogenesis of Taiwanese *G. parasuis* isolates.

## INTRODUCTION

*Haemophilus parasuis* (*H. parasuis*) is the causative agent of Glässer's disease, which induces sudden death, polyserositis, polyarthritis, meningitis, pneumonia and poor production performance, causing severe economic losses in the swine industry (*Zhang et al., 2014*). *H. parasuis* has been recently renamed in the NCBI tazonomy as *Glaesserella parasuis* (*G. parasuis*) under the family *Pasteurellaceae* (*Inzana, Dickerman & Bandara, 2016*). *G. parasuis* is commonly found in the porcine upper respiratory tract as part of the resident microbiota (*Moller & Kilian, 1990*). Previous studies have shown that multiple strains can be isolated from individual pig farms, even from single animals (*Lin et al., 2018*; *Turni & Blackall, 2010*). Therefore, accurate and effective typing techniques are required for identifying *G. parasuis* isolates. The isolates of *G. parasuis* have been classified into 15 serovars via classical immunological methods, including gel immuno-diffusion assays and indirect hemagglutination assays, but approximately 15–40% of pathogenic isolates are non-typable (NT) isolates, with NT rates varying by country or region (*Del Rio, Gutierrez & Rodriguez Ferri, 2003*; *Kielstein & Rapp-Gabrielson, 1992*). A molecular serotyping assay based on genetic differences among the capsule loci of the 15 serovar reference strains was developed to differentiate 14 of the 15 serovars of *G. parasuis* including NT isolates (*Howell et al., 2015*). Nevertheless, a variety of *G. parasuis* isolates belonging to the same serovar exhibit varying levels of virulence (*Aragon et al., 2010*; *Olvera, Segales & Aragon, 2007*). Non-typable isolates were also found in Taiwan using molecular serotyping (*Lin et al., 2018*). Although serotyping is useful for vaccine development and vaccine-based strategies, this method is not discriminative enough for differentiating between *G. parasuis* isolates.

One of the genotyping methods developed to assess the heterogeneity of *G. parasuis* isolates is enterobacterial repetitive intergenic consensus polymerase chain reaction (ERIC-PCR), which determines the relationships of *G. parasuis* isolates by comparing genetic fingerprints (*Oliveira, Blackall & Pijoan, 2003*; *Rafiee et al., 2000*). Compared to other genotyping assays, such as restriction fragment length polymorphism, multilocus sequence typing and single-locus sequence typing, ERIC-PCR has higher discriminatory power and lower cost (*Jablonski et al., 2011*; *Olvera, Segales & Aragon, 2007*).

Virulence-associated trimeric autotransporters (vtaA), belonging to the type Vc secretion system, which are divided into three groups may contribute to the adhesive/invasive capacity of *G. parasuis* (*Olvera et al., 2010*). Group 3 vtaA genes are highly conserved among virulent and non-virulent *G. parasuis* isolates, and group 1 vtaA genes are a potential indicator of virulent isolates (*Olvera et al., 2012*), making these useful targets in this present study. *G. parasuis* is a common pathogen that causes economic losses but no published research has been performed to assess the genetic lineages of *G. parasuis* isolates in Taiwan. Therefore, the aim of this present study was to characterize Taiwanese *G. parasuis* isolates via molecular serotyping, vtaA PCR, and ERIC-PCR.

## MATERIALS AND METHODS

### Vertebrate animal study

The study did not involve any animal experiment. This paper is an extension of a previous study (Lin et al., 2018). The Institutional Animal Care and Use Committee (IACUC) of National Pingtung University of Science and Technology did not deem it necessary for this research group to obtain formal approval to conduct this study.

### Bacterial isolates and culture conditions

A total of 145 *G. parasuis* strains were collected from 302 lesions of 166 diseased pigs. The animals were distributed across 130 diseased-pig herds on 97 pig farms (File S1); samples were collected between November 2013 and March 2017 in Taiwan. Lesions suspected of being caused by *G. parasuis*, including lesions in the meninges, pleura, pericardia, peritonea, synovial cavities and lungs, were examined by culturing on chocolate agar (at 37 °C, in 5% $CO_2$, and for 18–72 h depending on growth-rate variation among the isolates).

The bacterial isolates were identified by colony morphology, Gram staining (gram-negative bacillus), vtaA group 3 and molecular serotyping species-specific PCR results (Howell et al., 2015; Pina et al., 2009). *G. parasuis* isolates were stored at −80 °C. A loopful of bacteria from a passaged plate of pure culture was resuspended in 30 μL of ultrapure $H_2O$, which was heated to 100 °C for 30 min and centrifuged at $4,000×g$ for 1 min. The supernatant was used in the PCR.

### Pathological examination

Sick animals or fresh, complete carcasses were subjected to necropsy for gross morphological examination and H&E staining. Histopathological examination focused primarily on meningeal, pleural, pericardial, peritoneal and synovial cavities of joints and lungs. Pathological lesion patterns were determined to respiratory lesion, serositis of Glässer's disease and both lesions as in a previous study (Lin et al., 2018).

### ERIC-PCR assays

Enterobacterial repetitive intergenic consensus polymerase chain reaction was performed using an ABI 2720 (Applied Biosystems, Carlsbad, CA, USA) and a protocol that was modified from a previously published method (Dijkman et al., 2012). Each PCR was performed in a total volume of 30 μL containing ultrapure $H_2O$, 1× DreamTaq buffer, 250 μM dNTP, 0.17 μM primers ERIC-1F (5′-ATGTAAGCTCCTGGGGATTCAC-3′) and ERIC-2R (5′-ATGTAAGCTCCTGGGGATTCAC-3′), 1.25 U of DreamTaq DNA polymerase (Thermo Fisher Scientific, Waltham, MA, USA) and one μL of supernatant. The thermocycling conditions were as follows: 5 min at 94 °C; 40 cycles of 30 sec at 94 °C, 1 min at 50 °C and 2.5 min at 72 °C; and a final extension at 72 °C for 7 min.

The PCR products were stained with ethidium bromide and analyzed using a 20-cm-long 2% agarose gel. For electrophoresis, the gel was subjected to an electric field of six V/cm (300 V, 50-cm full-length electric field) for 3 h. A 50-bp DNA ladder RTU

(GeneDireX) and Bio-1D++ software (Vilber Lourmat, Collégien, France) were used to estimate molecular size.

## Molecular serotyping

Molecular serotyping assay for *G. parasuis* isolates was modified from a previously published method using the ABI 2720 (Applied Biosystems, Carlsbad, CA, USA) (*Howell et al., 2015*). An sp-sp amplicon was used as an internal control. Serotyping results of 279 isolates were from previous study (*Lin et al., 2018*).

## Virulence-associated trimeric autotransporters multiplex PCR (vtaA mPCR) assays

The mPCR protocol for group 1 and group 3 vtaA genes was modified from a previously published method using an ABI 2720 (Applied Biosystems, Carlsbad, CA, USA) (*Olvera et al., 2012*). Group 3 vtaA is highly conserved among invasive and non-invasive isolates, so this gene was employed for *G. parasuis* identification. Group 1 vtaA has been mainly detected only in virulent isolates (*Pina et al., 2009*). Polymerase chain reaction was performed in a total volume of 30 µL containing ultrapure $H_2O$; $1\times$ DreamTaq buffer; 250 µM dNTP; 0.17 µM YADAF1 (5′-TTTAGGTAAAGATAAGCAAGGAAATCC-3′), PADHR1 (5′-CCACACAAAACCTACCCCTCCTCC-3′), YADAF3 (5′-AATGGT AGCCAGTTGTATAATGTTGC-3′) and PADHR3 (5′-CCACTGTAATGCAATACCT GCACC-3′) primers; 1.25 U of DreamTaq DNA polymerase (Thermo Fisher Scientific, Waltham, MA, USA) and one µL of supernatant. The thermocycling conditions were as follows: 5 min at 94 °C; 35 cycles of 30 sat 94 °C, 30 s at 60 °C and 1 min at 72 °C; and a final extension at 72 °C for 7 min. The PCR products were stained with ethidium bromide and analyzed using a 2% agarose gel.

## Data analysis

The *G. parasuis* fingerprints were compared and analyzed using the unweighted pair group method with arithmetic mean with a 5% homology coefficient by using Bio-1D++ software (Vilber Lourmat, Collégien, France) to define strains. The cluster analysis was performed at a genetic similarity of 80%.

# RESULTS

## Genotyping of *G. parasuis* by ERIC-PCR

Three hundred and two isolates were identified as 145 strains, named strain 1–145, by same patterns of ERIC-PCR from same infected herds. Eighteen strains (12.4%) could not be typed using molecular serotyping due to absence of any serovar specific amplicon (abbreviated as NT). The results of the correlated clusters are depicted in Fig. 1. The ERIC-PCR dendrogram revealed a high level of genetic heterogeneity in many clusters. Two major clusters including 118 strains (81.4%) were observed in the dendrogram. Only 10 strains (6.9%) were clustered in five identical genotypes.

Upon applying an 80% similarity, 37 clusters were obtained. Partial correlation between the ERIC-PCR clusters of different strains, serovars and lesion patterns was observed.

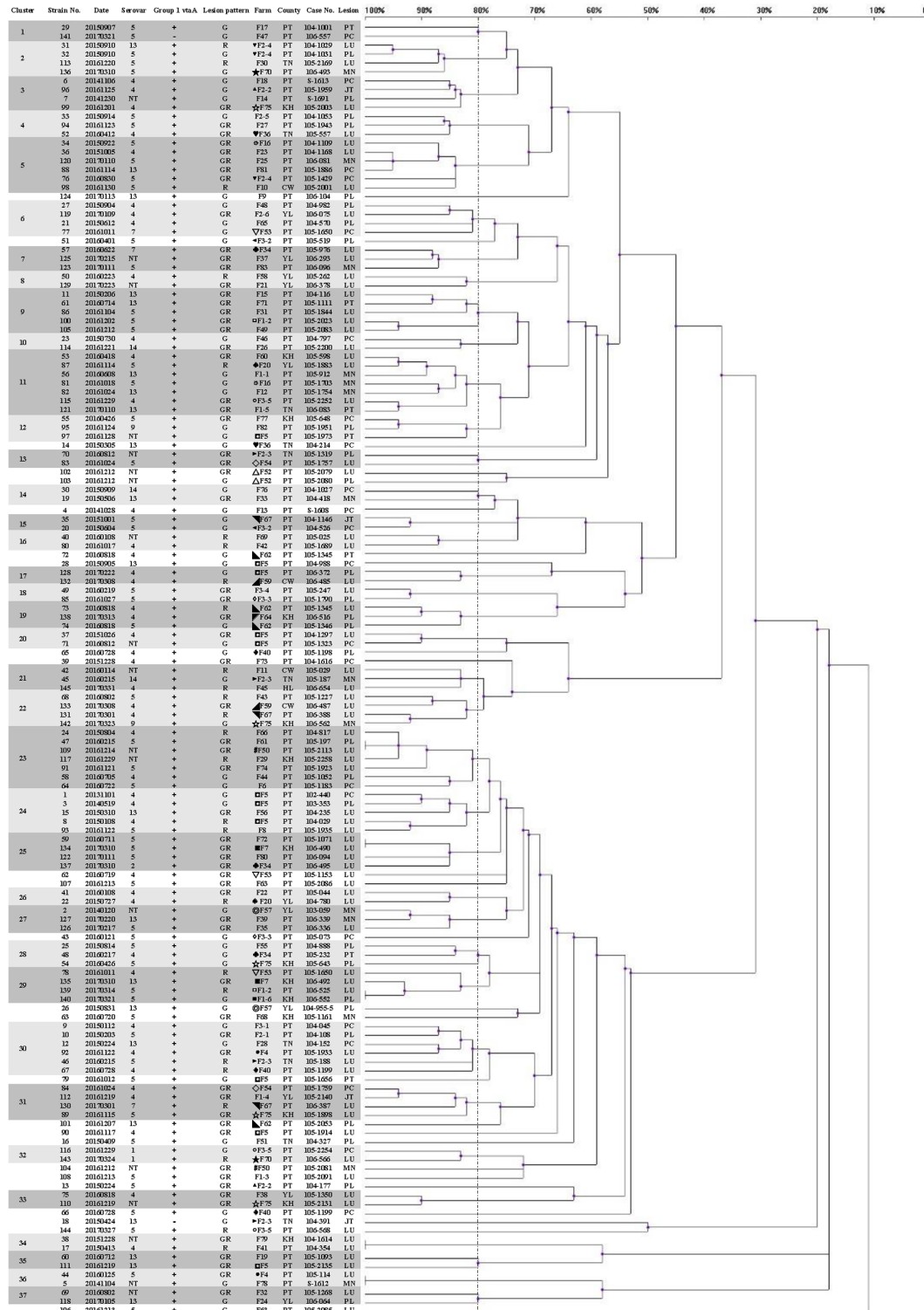

**Figure 1 Dendrogram showing relationships of and information on 145 *G. parasuis* strains.** Clinicopathological outcomes were divided into three types, namely, serositis of Glässer's disease (G), lower respiratory infection (R) and serositis with respiratory lesions (GR). County name abbreviations: Changhua (CW), Yunlin (YL), Tainan (TN), Kaohsiung (KH), Pingtung (PT), Hualien (HL). Lesions represent the origins of the strains. Lesion abbreviations: meninges (MN), pleura (PL), pericardium (PC), peritoneum (PT), joint (JT) and lung (LU). Different symbols represent strains from the same pig farm.

**Table 1 The relationship between strain, cluster and serovar in the same farms.**

| Farm | Strain no. | Cluster | Serovar |
|------|-----------|---------|---------|
| F5 | 1 | 24 | 4 |
| | 3 | 24 | 4 |
| | 8 | 24 | 4 |
| | 28 | None | 13 |
| | 37 | 20 | 4 |
| | 71 | 20 | NT |
| | 79 | None | 5 |
| | 90 | None | 4 |
| | 97 | 12 | NT |
| | 111 | 35 | 13 |
| | 128 | 17 | 4 |
| F75 | 54 | 28 | 5 |
| | 89 | 31 | 5 |
| | 99 | 3 | 4 |
| | 110 | 33 | NT |
| | 142 | 22 | 9 |

Some clusters included strains from one major serovar (proportion is greater than 70%), including clusters 1, 2, 3, 6, 15, 17, 18, 25, 26, 32 and 35. Some clusters included strains from one main lesion pattern (proportion is greater than 70%), including clusters 1, 3, 5, 6, 7, 9, 13, 15, 16, 18, 25, 28, 31, 33 and 35.

Strains 139 and 140, typed as an identical genotype and serovar 5, were isolated on similar dates and belonged to a same swine production system. Nevertheless, the lesion patterns of strains 139 and 140 were different. In farm F1-2, this strain was found to cause only respiratory lesions; however, this strain was found to induce polyserositis in farm F1-6 after one week. Although the nursery farms F1-2 and F1-6 are located in different counties, the weaning pigs were from the same sow farm. The spread of *G. parasuis* strains could be tracked according to the genotype.

The strains causing diseases on same farms at different time periods belonged to various clusters, like farms F5 (■) and F75 (☆). In farm F75, there were a total of five different strains belonging to different clusters in one year. Of total 11 strains in 5 years from farm F5, 2 and 3 strains belonged to clusters 20 and 24, respectively, but they were still different strains, even different serovars (Table 1).

From 12 herds (8.3%), there were 50 isolates (16.6%), identified as 27 strains (18.6%), which were isolated as more than one strain from same period endemic herds or same individuals. Some isolates were identified as different strains by both molecular serotyping and ERIC-PCR. Others isolates belonging to same serovars were identified as different strains via ERIC-PCR, namely, strains 65, 67, 72, 73, 74, 132 and 133. The others belonging to NT, strains 102, 103, were also differentiated by ERIC-PCR.

A panel of 33 isolates, identified as 24 strains by ERIC-PCR, was employed as a template to illustrate the relationship among isolates, strains, serovars, farms and infected organs

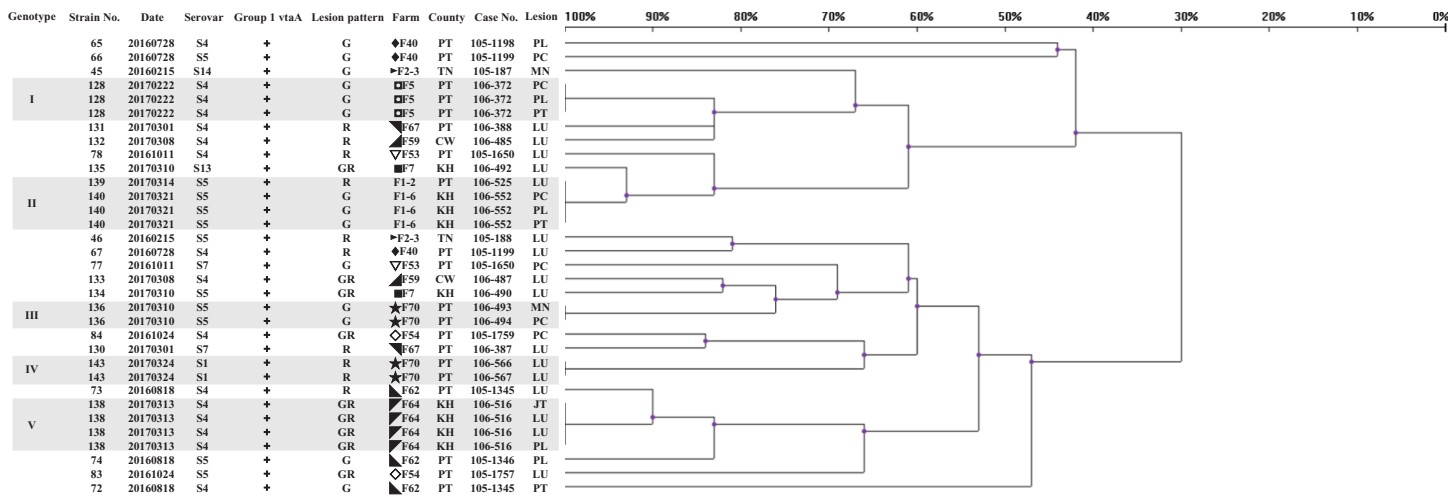

**Figure 2** Dendrogram showing relationships of and information on 33 *G. parasuis* isolates identified as 24 strains. The method of analysis and the meaning of the symbols are the same as in Fig. 1.

(Fig. 2). Most isolates from same animal or farm of origin were clustered in identical genotypes and typed within same serovar, namely, strains 128, 136, 138, 140 and 143 (data for other isolates were similar), and had homologous patterns (Figs. 2 and 3). The relationship between strains 139 and 140 was mentioned above. The other isolates from same period endemic herds or same individuals, as shown in Fig. 2, were determined to be heterogeneous via genotyping with or without serotyping.

## Virulence-associated trimeric autotransporters multiplex PCR

The group 1 and group 3 vtaA genes were detected via vtaA mPCR. A total of 145 strains were positive for group 3 vtaA (100%) and the molecular serotyping species-specific amplicon. Only two strains (1.4%) were negative for group 1 vtaA. One strain belonging to serovar 13 was isolated from the synovial cavity of an animal with pleuritis and arthritis. The other strain typed as serovar 5 was isolated from the pericardium of an animal with pleuritis and epicarditis. The positive rate of group 1 vtaA was 98.6%.

## Occurrence period of Glässer's disease

The age distribution of Glässer's disease was shown in Fig. 4. The highest occurrence period was from 5 to 8 week-old in nursery, followed by suckling period. However, there were still Glässer's disease cases in growing pigs and even in breeding period.

## DISCUSSION

To the authors' knowledge, this is the first study to describe the genetic diversity of *G. parasuis* in Taiwan. The ERIC-PCR dendrogram shows *G. parasuis* strains are genetically heterogeneous within the same serovar as well as between serovars. The great discriminatory power and typeability of ERIC-PCR are cited by previous studies of *G. parasuis* (*Dijkman et al., 2012*; *Jablonski et al., 2011*; *Moreno et al., 2011*; *Oliveira, Blackall & Pijoan, 2003*). In our study, ERIC-PCR is used to differentiate isolates within
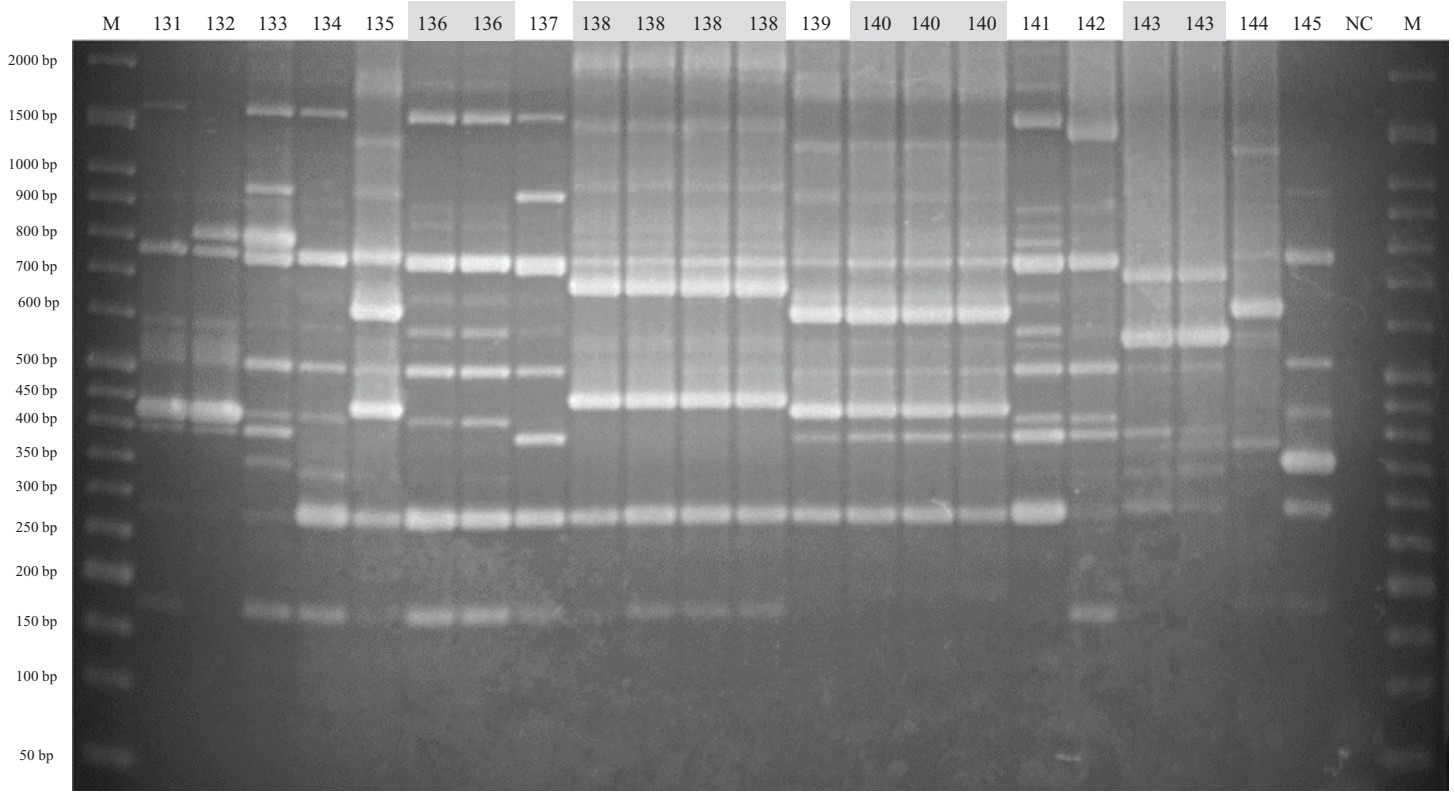

**Figure 3  ERIC-PCR gel image of *G. parasuis* isolates belonging to strains 131 to 145.** Lane M: 50-bp DNA Ladder RTU (GeneDireX); Lane NC: negative control.

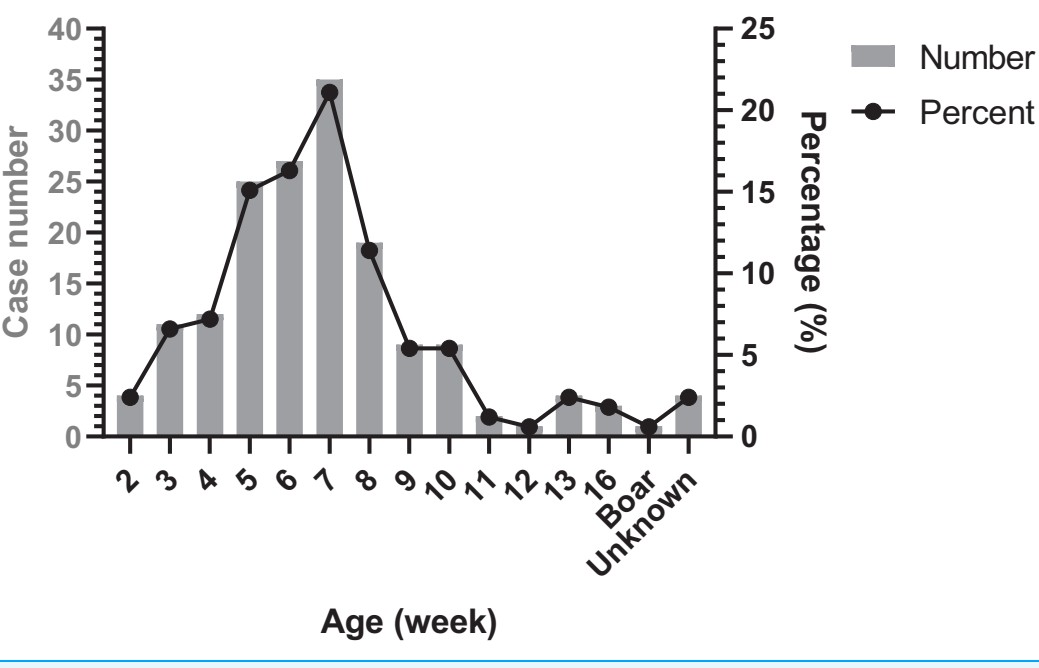

**Figure 4  Age distribution of diseased pigs.**

and between serovars, typed by molecular serotyping mPCR, which still remains molecular serotyping NT *G. parasuis* isolates in Taiwan (*Lin et al., 2018*). Therefore, the utility of ERIC-PCR for epidemiological survey is greater than that of not only immunological serotyping assays but also molecular serotyping assays.

In our data of strains from diseased animals, partial relationship is observed between genotypes by ERIC-PCR and serovars by molecular serotyping but the correlation is not obligate. Previous reports have indicated that the serovars are based on differences in the capsule genes of *G. parasuis* (*Howell et al., 2013*). Nonetheless, ERIC-PCR patterns can be obtained from the whole bacterial genome (*Olvera, Segales & Aragon, 2007*). The gap between these two typing assays may be caused by the scope of gene detection. Some strains isolated from different farms and at different times belong to the same serovars typed in the same clusters, which was likely due to homologous whole bacterial genomes. This may be related to the dissemination and evolution of *G. parasuis*. Additional data, such as data regarding antibiotic resistance and virulence genes, are worth collecting to better elaborate the epidemiology of *G. parasuis* for disease prediction, investigation, prevention and control in the future.

Cross-protection between strains belonging to the same serovar has been evidenced by several studies (*Miniats, Smart & Ewert, 1991*; *Smart & Miniats, 1989*; *Takahashi et al., 2001*). Nonetheless, heterologous protection across strains and serovars has been reported as being sporadic and inconsistent (*Oliveira & Pijoan, 2004*). Autogenous bacterins are considered an effective strategy against Glässer's disease (*Smart, Hurnik & Macinnes, 1993*). In this case, ERIC-PCR, which has great discriminatory power, can be a useful tool for isolate identification.

In Taiwan, porcine multiple-site production models are more popular. Many nursery farms, which have been modified from farrow-to-finish facilities, are owned by agricultural corporations and obtain weaning pigs from same sow farms. This phenomenon is common in Asian countries of which swine industries are developing. Sows are reservoirs of *G. parasuis* and transmit *G. parasuis* to suckling pigs (*Nedbalcova et al., 2011*). The results show pigs from two nursery pig farms sharing a sow farm are infected with the same *G. parasuis* strain at the same time. In addition, maternal immunity is an important factor influencing outbreaks of Glässer's disease (*Nedbalcova et al., 2011*). Furthermore, some swine production model usually mixes weaning pigs from different origins. Therefore, epidemiological monitoring of multiple-site pig farms and establishing herd immunity against *G. parasuis* strains is important for swine production systems for the prevention and control of Glässer's disease. Previous study indicates sows are reservoirs and passive antibodies can prevent Glässer's disease (*Nedbalcova et al., 2011*). Vaccination of sows will be an efficacious strategy to control Glässer's disease in suckling and weaning pigs. However, maternal antibody cannot protect late nursery and growing pigs. Vaccination of piglets can be considered to decrease economic losses caused by death and feed efficiency reduction. Our data showed various genotypes of *G. parasuis* causing Glässer's disease on the same farms at different time periods or same period endemic herds. Virulence varying within same serovars is caused by genome differences

and existence of virulence genes. The cross-protection among different genotypes belonging to same serovars should be concerned. Distinct clinicopathological outcomes in different farms caused by identical *G. parasuis* strains may be caused by various factors, such as room temperature, ventilation and coinfection with other pathogens.

It is well established that various strains can be isolated from individual pig farms, even from single animals (*Cerda-Cuellar et al., 2010*; *Oliveira, Blackall & Pijoan, 2003*; *Olvera, Calsamiglia & Aragon, 2006a*; *Olvera, Cerda-Cuellar & Aragon, 2006b*; *Ruiz et al., 2001*). ERIC-PCR can be used to identify isolates to assist *G. parasuis* recovery in challenge trials, sample collection for antibiotic resistance investigations, determination of antibiotic strategies for diseases causing by more than one strain and other studies requiring great discriminatory power. Although ERIC-PCR cannot offer information as valuable as whole-genome sequence data and comparisons of results among different laboratories is problematic, ERIC-PCR is still a useful, inexpensive, fast and practical tool for epidemiological surveillance, and in the identification of the source of transmission as an initial screening genotyping method. In the future, after typing by ERIC-PCR, molecular serotyping NT isolates are worthy of sequence analyses of polysaccharide biosynthesis loci. Based on sequence data and serovars by immunological serotyping assays, the gene functions, cross-protection among different serovars and new molecular serotyping assays can be investigated. In our study, group 1 vtaA are common in *G. parasuis* isolates from Taiwanese diseased pigs. This result is similar to the results of previous studies (*Dijkman et al., 2012*; *Moreno et al., 2011*; *Olvera et al., 2012*), so group 1 vtaA may also be important in the pathogenesis of Taiwanese *G. parasuis* isolates and worthy of developing as vaccine candidates. However, *G. parasuis* without group 1 vtaA still can be isolated from pigs with serositis (*Moreno et al., 2011*). Except host and environment factors, the pathogenicity of *G. parasuis* is considered to be related to multiple virulence factors (*Costa-Hurtado & Aragon, 2013*). Therefore, *G. parasuis* without group 1 vtaA may have ability to induce disease if other virulence genes exist and/or animals are immunosuppressive.

## CONCLUSIONS

This study shows the high genetic diversity of *G. parasuis* isolates in Taiwan. Enterobacterial repetitive intergenic consensus polymerase chain reaction has better discriminatory capability than molecular serotyping assays for the detection of slight differences among different strains and can be applied as an effective tool in the routine surveillance of *G. parasuis* for prevention and control strategies and further sequence studies. Group 1 vtaA are common in *G. parasuis* isolates from diseased pigs and may play important roles in the pathogenesis of Taiwanese *G. parasuis* isolates.

## ACKNOWLEDGEMENTS

The authors appreciate the help provided by the following people for this study: swine veterinarians at the Animal Disease Diagnostic Center for assistance with sample collection, Qiong-Yi Huang for assistance with bacterial preservation and Professor Shih-Chu Chen for providing the analysis software.

## ADDITIONAL INFORMATION AND DECLARATION

### Funding

The authors received no funding for this work.

### Competing Interests

The authors declare that they have no competing interests.

### Author Contributions

- Wei-Hao Lin conceived and designed the experiments, performed the experiments, analyzed the data, prepared figures and/or tables, authored or reviewed drafts of the paper.
- Hsing-Chun Shih analyzed the data.
- Chuen-Fu Lin contributed reagents/materials/analysis tools.
- Cheng-Yao Yang contributed reagents/materials/analysis tools.
- Chao-Nan Lin conceived and designed the experiments, contributed reagents/materials/analysis tools, prepared figures and/or tables, authored or reviewed drafts of the paper, approved the final draft.
- Ming-Tang Chiou conceived and designed the experiments, contributed reagents/materials/analysis tools, prepared figures and/or tables, authored or reviewed drafts of the paper, approved the final draft.

### Animal Ethics

The following information was supplied relating to ethical approvals (i.e., approving body and any reference numbers):

The study did not involve any animal experiment. This paper is an extension of a previous study (Lin et al., 2018. Molecular serotyping of *H. parasuis* isolated from diseased pigs and the relationship between serovars and pathological patterns in Taiwan. *PeerJ* 6:e6017 DOI 10.7717/peerj.6017). The IACUC of National Pingtung University of Science and Technology did not deem it necessary for this research group to obtain formal approval to conduct this study.

### Data Availability

The raw data are available in the Supplemental File.

### Supplemental Information

Supplemental information for this article can be found online at http://dx.doi.org/10.7717/peerj.6960#supplemental-information.

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
