# Peer review of "Genotypic analyses and virulence characterization of Glaesserella parasuis isolates from Taiwan"

_PeerJ, doi:10.7717/peerj.6960_

## Round 0.1 · original submission · Minor Revisions

As you revise your manuscript, while you do not need to change it to a short communication (PeerJ does not support alternative formats that vary according to paper length), you should keep the comments of each reviewer in mind, and respond to each point in your resubmission. In particular, please address the comments of reviewer #2 that related to using PCR to provide species identification.

Reviewer 1 ·

Basic reporting

The article is clear, figures are relevant and well described and it includes sufficient introdution.

The only thing that from line 153 to 166 the results section is not very clear, and it would have to be explained better.

Experimental design

Methods described with sufficient detail and it seems a rigorous investigation.

Validity of the findings

The work is interresting. The results provide very good information to be able to differentiate pathogenic isolates from those that are not.

Additional comments

From my point of view it is a good job that provides a lot of information about the characterization of isolates. Something that I would clarify better for the publication of the article in a magazine:

In the introduction, in line 53, you talk about microflora when the correct term would be microbiota.

In material and methods, line 87, do not specify how that selection of farms or at least how they were distributed was made. This information seems to me intersante, being able to even attach a map.

In line 88 it would be interesting to know if lesions were observed in the animals' encephalon.

In the discussion a lot of emphasis is made, even repeatedly that the same strain appears at the same time in different places and its relationship with the mothers and it is true that it is interesting. But I miss a deeper discussion about the possession of group 1 VtaA, for example, have really looked for these autotransporters in healthy animals to make a comparison. Because what is deduced from this study is that almost all isolates are pathogenic when G. parasuis is undesired as a diner as well.

The isolates studied would be by the techniques described in the well-characterized article but a more complete profile on these isolates with other virulence factors or even antibiotic resistance would complete the study.

Reviewer 2 ·

Basic reporting

Genotypic analysis and virulence characterization of Glaesserella parasuis isolates from Taiwan.

This is a nice study, however, it should be written as a short communication, as it is only the dendrogram that is discussed. The discussion is very long and goes over well-established knowledge. Some of the figures are not helpful in the discussion of the results and a couple of tables would be much better.
Abstract
Line 36 it should say within many clusters

Introduction

Line 52 Watt et all are talking about a new species which is in the process of being named. They have nothing to do with renaming Haemophilus parasuis. This was done by Thomas Inzana at a conference in Prato. Thomas is in the process of publishing this work in the taxonomic journal.
So the sentence should read that H. parasuis has recently been renamed as Glaesserella parasuis
The reference is Inzana, T.J., Dickerman, A.W. and Bandara, A.B. (2016) Taxonomic reclassification of “Haemophilus parasuis” to Glaesserella parasuis gen. nov., comb. nov. The Prato Conference on the Pathogenesis of Bacterial Infections of Animals 2016, Prato, Italy.
Line 55 Two references for this statement would be more than enough.
Turni and Blackall 2010 Serovar profiling of Haemophilus parasuis on Australian farms by sampling live pigs. Australian veterinary Journal 88, 255 – 259.
talk of up to 8 serovars on farm. Then you need one of the references that talks about more than one serovar per pig.

Materials and Methods
Line 94 says that bacterial isolates were identified as G. parasuis by the timeric autotransporter group. This is not a species specific PCR and the conditions are not geared at specificity. Some authors have argued that even the VtaA group 1 has problems and the conditions of the PCR to detect the gene were vital to specificity. The fact that you are using a tiny amount of water (normally 100 to 200 ul are used for one loopful to obtain about 100 ng of DNA) to extract DNA is a worry. The study suggested that previously positive results for non-pathogenic strains were related to the conditions of the PCR not being exactly followed, as some non-pathogenic strains have a very large trimeric autotransporter whose translocator sequence is characteristic of group 1 vtaA. (Galofre-Mila N, Correa-Fiz F, Lacouture S et al. A robust PCR for the differentiation of potential virulent strains of Haemophilus parasuis. BMC Vet Res 2017;13:124.)

You are doing the serovar PCR that has a species specific band. Why did you not go by these results to determine if it is G. parasuis? It should state somewhere what the species specific outcome of the serotyping PCR was. As the vtaA group 3 is not a convincing result for species.

Line 101 remove the coma before the and.
Line 103 as in a previous study
Line 140 which city and country does this software come from?

Results
Figure 1 What are all these symbols in the farm column?
Can you take out the S from the serovars, as it is very hard to read? The case number is of no value to the paper, I would take this out. The date should only be the year.

Why would you apply a 35% similarity cut-off. I find your 80% similarity cut off too liberal. Most authors use a 90% and higher cut off. (Characterization of the diversity of Haemophilus parasuis field isolates by use of serotyping and genotyping. Oliveira S, Blackall PJ, Pijoan C Am J Vet Res. 2003 Apr; 64(4):435-42.) (Short Communication: ERIC-PCR genotyping of Haemophilus parasuis isolates from Brazilian pigs. Nubia ResendeMacedo,Simone RodriguesOliveira Andrey PereiraLage José LúcioSantoscMarina RiosAraújo Roberto Maurício CarvalhoGuedes The Veterinary Journal Volume 188, Issue 3, June 2011, Pages 362-364
Line 148 I do not understand why we need to show a 35% similarity cut – off . I would remove this section.

I would re-evaluate the whole dendogram with a cut off of 90%. This would make your results comparable to the rest of the studies published already.

I would have started the results with the serovars observed. There seems to be only very few serovars that have been found. The diversity is much more interesting in the light of the few serovars found. Your vtaA comes up positive for all serovars even serovar 7 and 9, both of which are in general non-pathogenic with one exception for a serovar 5.
Line 154 Are you refereeing to the major serovar being 5. Can you please spell out the serovar, as you have not explained the serovars before. But why is subtype 4 and 5 not included for example?

Line 155 you state subtype 5 as having isolates from the same lesion pattern, but that first mentioned patterns has lesion pattern R and GR in it. So it is mainly the same pattern, but not the same pattern. 13 and 14 are missing in that list.

There need to be some paragraphs when we change topic. Such as a different paragraph at line 150 and at 155 when isolates were discussed and 160

The results are extremely difficult to follow as the writing in you dendogram is way to small.

A table such as below for lines 160
Farm subtype serovar
F5
F75

Give us some examples of isolates from the same period from endemic herds and the different genotypes and of the different isolates by both subtype and serovar.

Figure 2 is not helpful as you are not showing us in the table this pattern you referring to. A table would have been much more helpful. Figure 3 is not needed. Figure 2 is not readable and a table would be much better to express all the data.

Discussion
Line 197 You have to be careful you only had very few serovars and correlation between small groups is easy then, but might not be real.
Line 204 This might be due to dissemination and could be due to coming from the same breeding stock originally. Not sure if it has to do with the evolution of G. parasuis.
Line 230 on the same farms
Line 231 and existence of virulence genes.
Line 236 It is well established that various isolates can be isolated……

Experimental design

There is nothing wrong with the design. However, the species needs to be identified with a species specific PCR. The cut off for the Dendrogram needs to be 90% or higher.

Validity of the findings

Data needs to be reevaluated with the different cut off

Additional comments

Please see comments in the first area

---

## Round 0.2 · Minor Revisions

As indicated above, there is an excellent chance we will accept your manuscript for publication. As a final step, please review the suggestions and comments from reviewer #2, and consider modifying your manuscript accordingly.

Thank-you

Reviewer 1 ·

Basic reporting

The work exposes interresting results and throws new ways of investigation with this bacterium. It is a professional article structure, figures and tables.

Experimental design

The investigation must have been conduced rigorously and to a high technical standard.

Methods described with sufficient detail and information to replicate

Validity of the findings

The results are significantly rigorous, with their replicas and their statistical treatments. The conclusions are the right ones.

Additional comments

After the first review, the article is read better and the main ideas are clear. From my point of view the development of the work is well explained and the conclusions obtained are optimal to consider this work as valid for publication in this journal.

Reviewer 2 ·

Basic reporting

This is a nice report and has now addressed some of the major flaws

Experimental design

Well designed

Validity of the findings

Findings are valid

Additional comments

Thank you for addressing the points. I have pointed out some minor things still which in my opinion need to be addressed. I have also addressed some of the points argued in the letter of response.

Looking at the figure 1 and 2 and at the cut-off point it becomes clear that the cut off point is 90%. At 90% your isolates have the same genotype and a clonal. When looking at your genotypes then it becomes clear that the 90% gives you that answer. Looking at the figure 2 and 3 you can see that isolate 135 is very similar to 139 and 140 and could belong in the same cluster. When you are extending this to the 80% cutoff then you get 133 being the same as 134. These are definitely not the same on the contrary they are quite different. So contrary what you wrote before there are definitely not only 37 subtypes. However seeing that you now changed that into clusters then I am not objecting to your analysis of 80%. A subtype is a different strain and not a cluster. So you have 37 clusters.
Seeing that you identified the field isolates beforehand at 95% similarity assures that you had representatives of all the isolates.
At line 155 we talk of 36 clusters yet 37 are in the table.
In line 169 to 173 you are talking about genotypes and calling the cluster a genotype. The genotype is the subtype/strain. So there are more subtypes in the cluster as already discussed. So you can only discuss the clusters here and your table was labelled correctly, but your text is incorrect.

I beg to differ with you argument in the corresponding letter, as we have seen that Glaesserella parasuis is a very diverse pathogen and when Australian strains were added to the MLST database they were all different to existing strains from Spain and the US in the data base. ERIC is representing this variation. The discriminatory power of 80% is definitely not representative of the variation and the profile is quite different when you look at the similarity by eye. ERIC for H. parasuis is very consistent and you will get the same pattern when run on different occasions with some serovars having distinct bands. You again mixing up clusters with subtypes.
The cut off is meant to give you the same strain (subtype) when above the cut off. Below you get different strains. The same farm can have different strains with the same serovar. A study by Turni et a. (2010) revealed that on some farms, more than one genotype was present for one serovar.
What you are looking at is clusters of related strains and for that you can have the 80% cut-off

An isolate is a non-defined isolate and the differentiation between field isolate and isolate is strange. The figure just shows which isolates are the same and therefore one strain.
It would have been good to discuss the fact that all bar two isolates had the vta gene, as other studies have not shown that.
The ERIC PCR can not necessarily detect different serovars. There are cases where the ERIC PCR is the same but the serovar is different. That might have to do with the change in serovar might not be due to huge changes in the DNA and therefore might not be noticeable in the distance of the repetitive sequences and the banding pattern might stay the same. This is particularly relevant for serovar 12 and 5, which even the new multiplex PCR can not hold apart.

---

## Round 0.3 · accepted · Accept

Thank-you for addressing the comments of the reviewers. I am happy to be able to accept your manuscript for publication.

#